# Physical and Mechanical Performance of Coir Fiber-Reinforced Rendering Mortars

**DOI:** 10.3390/ma14040823

**Published:** 2021-02-09

**Authors:** Cinthia Maia Pederneiras, Rosário Veiga, Jorge de Brito

**Affiliations:** 1CERIS, Instituto Superior Técnico, University of Lisbon, 1049-001 Lisbon, Portugal; cinthiamaia@tecnico.ulisboa.pt; 2National Laboratory for Civil Engineering, 1700-066 Lisbon, Portugal; rveiga@lnec.pt

**Keywords:** vegetable fiber, fiber-reinforced mortar, cement and cement-lime mortars, sustainability, render

## Abstract

Coir fiber is a by-product waste generated in large scale. Considering that most of these wastes do not have a proper disposal, several applications to coir fibers in engineering have been investigated in order to provide a suitable use, since coir fibers have interesting properties, namely high tensile strength, high elongation at break, low modulus of elasticity, and high abrasion resistance. Currently, coir fiber is widely used in concrete, roofing, boards and panels. Nonetheless, only a few studies are focused on the incorporation of coir fibers in rendering mortars. This work investigates the feasibility to incorporate coir fibers in rendering mortars with two different binders. A cement CEM II/B-L 32.5 N was used at 1:4 volumetric cement to aggregate ratio. Cement and air-lime CL80-S were used at a volumetric ratio of 1:1:6, with coir fibers were produced with 1.5 cm and 3.0 cm long fibers and added at 10% and 20% by total mortar volume. Physical and mechanical properties of the coir fiber-reinforced mortars were discussed. The addition of coir fibers reduced the workability of the mortars, requiring more water that affected the hardened properties of the mortars. The modulus of elasticity and the compressive strength of the mortars with coir fibers decreased with increase in fiber volume fraction and length. Coir fiber’s incorporation improved the flexural strength and the fracture toughness of the mortars. The results emphasize that the cement-air-lime based mortars presented a better post-peak behavior than that of the cementitious mortars. These results indicate that the use of coir fibers in rendering mortars presents a potential technical and sustainable feasibility for reinforcement of cement and cement-air-lime mortars.

## 1. Introduction

Agricultural waste has been considered an environmental issue. Coir fiber is a by-product waste of the production of other coconut products [1,2], and the world production is approximately 250,000 tonnes a year [3]. In order to provide a proper disposal, many researchers seek different approaches to use the coir waste fiber. Concerning engineering applications, coir has been incorporated in concrete, roofing, boards, panels, and others building materials [4,5,6,7,8,9,10]. Coir fibers are extracted from between the outer husk of coconut and the internal shell [11], and their physical and mechanical properties are seen as great potential to improve the ductility, flexural toughness, and energy absorption capacity of the composites. The high toughness and flexibility of these fibers offer a better post-cracking behavior of the reinforced composites.

Regarding the incorporation of coir fibers in mortars, a few studies were carried out with the purpose of enhance their cracking performance [1,12,13,14,15,16,17]. In previous studies, the use of coir fiber was investigated in mortars with cement as the only binder (further referred as cement mortars) and in mortars using more than one binder. In general, the authors have reported that the addition of coir fibers decreased the workability of the mortars; thus, it was necessary to add more mixing water when compared to that of the mortar without fibers. This effect is attributed to the high water absorption and the retentive nature of the coir fibers [6]. A reduction in the mortar’s density was found by increasing the coir fibers content and water/binder ratio.

Concerning the compressive strengths, there are contradictory outcomes presented in previous works. For cement mortars, Hwang et al. [1] found that the incorporation of coir fibers decreased their compressive strength when increasing the fiber content. The authors attributed this reduction to the fibers clustering inside the matrix; an increase of the volume of voids inside the composite was observed as a high volume fraction of coir fibers was added, which indicates a more porous structure. On the other hand, other researchers found an increase in compressive strength with the incorporation of coir fibers in cement mortars [12,15]. Al-Zubaidi [15] attributed this effect to the distribution of stresses by the fibers. In what concerns the addition of coir fiber in cement-lime mortars, Sathiparan et al. [14] found that the compressive strength of the mortars increased with the volume fraction of coir fibers up to 0.5%, whereas a higher content of coir fiber decreased the compressive strength of the mortar when compared to the reference mortar.

With regard to flexural properties, it is well known that the addition of fibers increases the flexural strength, fracture toughness, and ductility of the mortars. Hwang et al. [1] reported a significant improvement in flexural behavior of the coir fiber-reinforced cement-based mortars. The flexural strength increased by increasing the fiber content. The explanation given by the authors was that the fibers distribute the stresses before rupture. Additionally, the coir fibers surface seems rough, which provides a better interfacial adhesion between the fiber and the cementitious matrix. Andiç-Çakir et al. [12] also found an increase in flexural strength with increasing of the coir fiber amount in cement mortars.

For cement-lime mortars, the incorporation of coir fibers in the studies of Sathiparan et al. [14] presented improvements in flexural strength up to 0.5% of addition. The authors reported that the incorporation of 0.75% coir fiber decreased by 16.5% the flexural strength when compared to the reference mortar.

Previous studies have found an enhancement in mortar’s toughness as increasing the addition of coir fibers [1,12,14]. Sathiparan et al. [14] reported that the flexural toughness indices of the cement-lime mortars were significantly higher than that of the control mortar, which reveals a higher energy absorption during post-peak. The authors calculated the toughness indices based on the total area under the load-deflection curves from the flexural strength test. Therefore, the authors verified that the flexural ductility of the mortars with coir fibers have increased by increasing the coir fiber content, i.e., the fiber-reinforced mortars showed a more ductile failure mode when compared to that of the mortars without fibers. Regarding the incorporation of coir fibers in cement-based mortars, they also presented improvements in terms of residual strength, ductility and toughness. Hwang et al. [1] found that the increase of coir fibers content presented a significant increase in mortars’ toughness. This improvement was associated to the bridging mechanism of the fibers, which transfers the stress in the matrix across the opening cracks and withstands a residual load after achieving the maximum load. The results found by Andiç-Çakir et al. [12] also presented a remarkable increase in toughness values, which can be attributed to the bridge phenomena and interface between fibers and the matrix.

Cracking behavior of the mortars with vegetable fibers was also investigated in previous works [1,12,14,16]. It was clear that the effectiveness of the fibers in led to reducing the mortars’ shrinkage. Toledo Filho et al. [16] evaluated the free, restrained, and drying shrinkage of coir fiber-reinforced cement mortars. The authors reported that the addition of vegetable fibers delayed the first crack opening and crack propagation in the mortars. This effect is mainly due to the bridging mechanism of the fibers across the cracks. Hwang et al. [1] and Sathiparan et al. [14] also found that the incorporation of coir fibers contributes to control the cracking opening and its propagation, due to the stresses distribution by the fibers. As a result, conversely to the mortars without fibers that present a brittle failure, the modified mortars present a ductile behavior and a gradual failure.

Notwithstanding these previous research studies, the study of coir fibers in rendering mortars with two different binders has not been found in the literature. Therefore, the aim of this work was to investigate the feasibility of the renders with the addition of coir fibers, in order to minimize cracking by improving the mortars’ ductility. Cement and cement-lime mortars with compositions adequate for use as renders were produced and modified by adding 10% and 20% of coir fibers by the total mortar volume. Two common volumetric ratios were used to produce the mortars for render’s application following the European Standard for specifications for rendering mortars (EN 998-1 [18]). Cement to aggregate ratio at 1:4 and cement: air-lime: sand at 1:1:6 volumetric ratio. Two fibers’ length were chosen based on previous research studies that used coconut fiber as reinforcement in mortars. The coir fiber-reinforced mortars’ properties were investigated at fresh and hardened state, and physical and mechanical behavior was evaluated through several tests.

## 2. Experimental Program

### 2.1. Materials

The objective of the experimental program was to evaluate the physical and mechanical properties of cement and cement-air-lime-based mortars with coir fibers for non-structural uses, namely renders. The materials used in this study are the following:Cement (Secil, Portugal): CEM II/B-L 32.5 N, according to EN 197-1 [19];Calcium hydrated lime powder-air lime (Calcidrata S.A., Portugal): Class CL80-S, according to EN 459-1 [20];Sand (Areipor—Areias Portuguesas S.A., Portugal): Sieved river sand to obtain the size range previously defined;Coir fibers (waste from an insulation company—Amorim Cork Insulation, Portugal): With lengths of 1.5 cm and 3.0 cm.

The results of the tensile properties of the coir fibers used in this current work are: tensile strength of 237.26 ± 79.55 MPa, modulus of elasticity of 2.25 ± 1.75 GPa and elongation at break of 11.25%. The coir fiber’s water absorption is 115%. Coir fibers consist of cellulose as crystalline microfibrils held together by amorphous lignin and hemicellulose fibrils [21]. In general, plant-based fibers present a similar morphology. The cellulose fibrils packed inside these bundles bonded with lignin forming an unidirectional filament [22].

The grading curve of the sand used in this work is presented in Figure 1. The sand was previously washed and calibrated by the producer. From the technical sheet of the producer, the sand is mainly composed by quartz (>98% silica). The sand was sieved to achieve the previously size distribution chosen. The opening of the sieves were 0.063, 0.15, 0.25, 0.50, 1.00, 1.70, and 2.00 mm.

The fibers’ length was obtained by manually cutting of the waste, which is presented in Figure 2. These fibers were previously washed with neutral detergent, in order to remove any impurities. Before the fibers incorporation in the mix, they were distributed inside a properly closed receptacle by blowing compressed air in order to achieve an adequate dispersion and disentangle the fibers.

A microscopic observation was performed using an Olympus SZH-10 optical microscope (Tokyo, Japan) in order to evaluate the fibers’ surface and estimate their diameter. Figure 3 presents a micrograph of coir fiber, and the average of coir fibers diameter is 179 ± 3 μm. Therefore, the aspect ratios (length/diameter) were 83.80 and 167.60 for 1.5 and 3.0 cm, respectively.

The bulk density of the constituents of the mortars produced is presented in Table 1.

### 2.2. Mix Design

The mortars were produced at two volumetric ratios: 1:4 (cement: aggregates) and 1:1:6 (cement: air-lime: aggregates). The water to binder ratio varied according to the amount of mixing water required in each mortar, since the consistency by flow table value was fixed at 140 ± 2 mm, which provides adequate workability for renderings. The composition of the mortars produced in this work is presented in Table 2. The incorporation of the fibers waste was analyzed at two ratios: 10% and 20% by total volume of the mortar; since the coir fiber bulk density is low, the contents expressed in volume represent a considerable weight of incorporation, as seen in Table 2. Two different lengths were used: 1.5 cm and 3.0 cm.

### 2.3. Methods

The standards and number of specimens used for each test performed are listed below. The properties determined in the fresh and hardened mortars tests were:Consistency of fresh mortar (by flow table)—EN 1015-3 [23]. Three samples per mortar.Bulk density of fresh mortar—EN 1015-6 [24]. Three samples per mortar.Dry bulk density of hardened mortar—EN 1015-10 [25], at 28, 90, 180, and 365 days. Three prisms per mortar.Flexural strength of hardened mortar—EN 1015-11 [26], at 28, 90, 180, and 365 days. Three prisms per mortar.Compressive strength of hardened mortar—EN 1015-11 [26], at 28, 90, 180, and 365 days. Six prisms per mortar.Dynamic modulus of elasticity by resonance frequency of hardened mortar—EN 14146 [27], at 28, 90, 180, and 365 days. Three prisms per mortar.Ultrasound pulse velocity of hardened mortar—EN 12504-4 [28]. To measure this property, two methods were applied: direct and indirect. In the direct method, the electrodes are on opposite sides of the prisms and, in the indirect method, the electrodes are on the same surface of the prisms. The direct method measures the wave’s propagation time between extremities and the indirect method makes the measurements in small increasing distances on the same surface. This test evaluates the mortar’s compactness; a lower wave propagation velocity indicates a less compact material, since it means a greater volume of intercepted voids. Three prisms per mortar at 28 days.Open porosity—EN 1936 [29]. Three samples per mortar, resulting from the compressive strength test at 28 and 365 days.

Prismatic samples with dimensions of 160 × 40 × 40 mm^3^ were used for the hardened mortars tests. The mortars were cured as specified by EN 1015-11 [26], which establishes that the specimens should be kept inside the molds for two days at a temperature of 20 ± 2 °C and a relative humidity of 95 ± 5%. Then, the prisms are demolded and the specimens kept in the same conditions for 5 days inside of plastic bags. After seven days of the mortars production, the specimens were kept in a room with temperature of 20 ± 2 °C and 65 ± 5% of relative humidity, until the day of the test.

## 3. Results and Discussion

### 3.1. Workability

The fresh behavior of the mortars was evaluated through the consistency by flow table test and bulk density. The workability was previously chosen by a proper consistency of the mortar to be applied on vertical surfaces. In order to ensure an adequate workability, an application on a brick was carried out. The consistency of the mortars was previously fixed at 140 ± 5 mm. Therefore, the amount of mixing water needed by each mortar was different. The water to binder ratios are presented in Table 3. It was noticed that the incorporation of coir fibers increased the water content required in order to achieve the intended workability. Nonetheless, it can be seen in Figure 3 that the fibers maintain the mortar agglutinated. The water/binder ratio increased by increasing the fibers length and volume fraction. C 3.0-20c presented the highest increase of about 12% compared to that of the REF 1:4. It is also stressed that longer fibers presented worse workability than the shorter ones. The finding of coir fibers addition in cement-based mortars reducing the workability was also presented by Hwang et al. [1] and Andiç-Çakir et al. [12]. The authors reported that as increasing the fibers volume fraction, the mortars workability decreases due to the fibers clustering.

Mortars with 10% of coir fibers using a mix of cement and air-lime as binders presented similar water/binder ratio to the mortar without fibers, i.e., the addition of fibers did not affect the mortars workability. On the other hand, the addition of 20% of coir fibers showed an increase of the water required when compared to that of the REF 1:1:6. The results found in this work are in agreement with the one found in the technical literature. Sathiparan et al. [14] noticed that, as increasing the coir fibers content in cement-lime mortars, the amount of water needed has increased.

The bulk density of the mortars reduced with the addition of coir fibers, regardless of the type of binder used. However, this reduction is more significant for the modified cement-based mortars. This could be due to the lower bulk density of the fibers. Figure 4 shows the bulk density test’s preparation.

In short, the incorporation of fibers reduced the mortars’ workability. Therefore, a higher mixing water content was used in order to reach an intended flow table value. The mortars with fibers showed an increase in water to binder ratio when compared to the reference mortar, which can affect the hardened properties of the mortars. It was noticed that the workability decreased as the fibers’ length and volume fraction increased.

### 3.2. Dry Bulk Density

Figure 5 presents the dry bulk density results of the mortars at 28, 90, 180, and 365 days. A similar trend of the fresh bulk density was observed. The addition of coir fibers reduced the bulk density of the mortars. It was found for C 3.0-10c a decrease of 4.5% in bulk density when compared to the reference mortar at 365 days. Regarding the cement-lime mortars, this reduction was not significant. Hwang et al. [1] also found a reduction in bulk density of the modified mortars, this effect was attributed to the low density of the natural fiber and the higher porosity of the modified mortars.

### 3.3. Dynamic Modulus of Elasticity 

The dynamic modulus of elasticity was determined by resonance frequency over time, from 28 days until 365 days, and the results are presented Figure 6. Rendering mortars should be able to deform and accommodate the stresses without cracking. The modulus of elasticity measures the mortar’s ability to deform under stress. A low modulus of elasticity indicates that the mortars present a certain deformability, which may prevent cracking. Regardless of the binder used, it was noticed that the incorporation of fibers reduced the modulus of elasticity, which means that the modified mortars may behave in a more deformable way than the control mortar.

At 365 days, cement-based mortar with 10% of 3.0 cm coir fibers presented the highest decrease of approximately 21% when compared to the reference mortar. In what concerns the cement-lime mortars, the fibers’ length and volume fraction seemed not to affect this reduction, since all the modified mortars presented similar values among them. The lowest modulus of elasticity was attributed to the C 1.5-10cl sample, which reduced approximately 19.5% compared to that of the REF 1:1:6, at 365 days.

A reduction in the modulus of elasticity with the addition of natural fibers in mortars was also found in the literature. For cement-based mortars, Maia et al. [30] reported a decrease in modulus of elasticity when natural sheep’s wool fibers were added. Sathiparan et al. [14] also noticed a reduction in modulus of elasticity when coir fibers were incorporated in cement-lime mortars. Therefore, the authors stated that the coir fiber-reinforced mortars presented a more ductile behavior than that of the control mortar.

### 3.4. Ultra-Sound Pulse Velocity

The ultra-sound pulse velocity of the mortars is presented in Figure 7, at 28 days. Two methods were used: direct and indirect. The ultra-sound pulse velocity results, in both methods, showed that the incorporation of fibers reduced the pulse velocity through the mortar. This result indicates that the modified mortars present a higher volume of pores and suggests a decrease of the mortars modulus of elasticity with the addition of the coir fibers, which is consistent with the results of the resonance frequency test.

A reduction in ultra-sound pulse velocity was also observed by Maia et al. [30]. The authors incorporated sheep’s wool fibers in cement and cement-lime mortars, and, regardless of the binder used, the natural fiber-reinforced mortars presented less compactness when compared to the reference mortar. These results corroborate the reduction in modulus of elasticity, since the modified mortars may deform more than the mortars without fibers.

### 3.5. Compressive and Flexural Strengths

Compressive and flexural strengths tests were performed at 28, 90, 180, and 365 days, and the results are presented in Figure 8 and Figure 9.

Rendering mortars should not exhibit a high compressive strength, which indicates high rigidity, since a brittle behavior may be more susceptible to cracking. Therefore, these results obtained in this work may be a positive factor for a render. On the other hand, the flexural strength is most requested and should withstand the building movements and thermal variations stresses without cracking. Flexural strength is strongly correlated to other characteristics, such as susceptibility to cracking and adhesive strength of rendering mortars.

For cement-based mortars, the incorporation of coir fibers slightly decreased the compressive strength, in general, with the exception of C 1.5-20c, which presented an increase in compressive strength when compared to the reference mortar until up to 180 days. C 1.5-10c presented the lowest compressive strength, which was a reduction of about 32% when compared to the reference mortar at 365 days. In terms of flexural strength, it can be noticed an increase in flexural strength with the coir addition until 90 days. After that, a reduction is evidenced for all the samples. C 1.5-10c had the major reduction of approximately 30% when compared to the REF 1:4, at 365 days. The volume fraction had more influence in the mechanical strengths than the length of the fibers, since C 1.5-20c and C 3.0-20c presented similar values at 28 and 365 days.

The mortars with cement and air-lime as a binary binder followed the same trend: the addition of coir fibers reduced the mortars’ compressive strength in the first ages. However, at 365 days, the C 3.0-10cl and C 1.5-20cl mortars both obtained an increase of 6% compared to the reference mortar. This effect could be due to the improvement in interfacial transition zone between the matrix and the fibers over time, which provides a better distribution of the stresses when submitted to loading.

The major reduction was presented by C 1.5-10cl, which showed a decrease of 12% in relation with the REF 1:1:6, at 365 days. Concerning the flexural strength, the incorporation of coir fibers increased when compared with a reference mortar, at 28 days. This improvement lasted up to 180 days. Modified mortars presented a reduction in flexural strength at 365 days, as increasing the fiber content and fiber length. C 3.0-20cl presented major reduction of 19% compared to the REF 1:1:6. It can be seen that the reduction over time is more relevant for cement-based mortars. This could be due to a higher water to binder ratio in cement-mortars with coir fibers and the fiber’s degradation over time. The use of air-lime in the binder improved the fibers performance in terms of flexural strength, since the cement-lime mortar with coir fibers required less water to achieve the intended workability than the reference mortar; thus, the cement-lime mortars presented higher improvements in mechanical strengths than the mortar without fibers.

Figure 10 presents the coir fiber-reinforced cement-based mortars specimens after compressive strength and flexural strength tests.

In short, the compressive strength of the coir fiber-reinforced mortars showed a decrease. C 1.5-10c and C 1.5-10cl had the lowest compressive strength, a reduction of 32% and 12% compared to the respective reference mortars. Concerning flexural strength, the use of coir fibers enhanced the flexural strength up to 90 days of the cement-based mortars. The flexural strength of the cement-lime mortars showed an improvement until 180 days. It was noticed that, over time, the flexural strength of the modified mortars presented similar values to those of the mortars without fibers. It is stressed that the coir fiber may degrade inside the matrices of cement or cement-lime mortar due to their composition. Therefore, a reduction in the improvements provided by the fibers over time was noticed. A treatment for coir fibers could enhance their performance.

The findings of previous studies present different results based on the fibers lengths and volume fraction. The results presented in the literature referred to 28 days tests. Hwang et al. [1] followed the trend found in the results obtained in this paper, i.e., the authors reported a reduction in compressive strength of cement-based mortars as the coir fibers content increased. Hwang et al. [1] and Andiç-Çakir et al. [12] found, similarly to this research, that the coir fibers incorporation increased the flexural strength of cement-based mortars.

Regarding the cement-lime mortars, Rupasinghe et al. [31] and Sathiparan et al. [14] found that the incorporation of coir fibers increased the compressive strength up to 0.5% of incorporation. Mortars with higher volume fraction reduced the compressive strength. The flexural strength of mortars with 0.5% of coir fraction was 6% higher than the control mortar, whilst the addition of 0.75% reduced the flexural strength around 16.5%. The authors attributed this reduction to the fiber’s clustering.

### 3.6. Cracking Behaviour

The cracking behavior of the mortars was evaluated through some parameters proposed by the Center Scientifique et Technique du Bâtiment (CSTB) [32] and by their fracture toughness, which are presented in Table 4. Rendering mortars should dissipate the tensile stresses without cracking, i.e., the mortar’s capacity to absorb and accommodate the tensions correlates to its cracking resistance. In order to analyze the cracking susceptibility of the mortars, these parameters may indicate the mortars’ ability to resist cracking. The former criterion is based on the ratio between the modulus of elasticity and flexural strength (E/σ_f_). It is well known that a low modulus of elasticity contributes to a better deformation capacity, and a high flexural strength indicates a better mechanical resistance to support the load applied. Consequently, when the E/σ_f_ ratio is high, the tendency of the mortar to crack is greater. The other factor is related to another ratio (σ_f_/σ_c_), which is between the flexural strength and compressive strength, which suggests the ductility of the material, i.e., when the σ_f_/σ_c_ ratio is closer to one, the mortar tends to be more ductile. The deformability of the mortar before failure is measured through the ductility of the material. The fracture toughness indicates the mortars ability to absorb energy during failure, and it can be measured through the area under the load-deflection curves from the results of flexural strength at 28 days [33]. These load-deflection curves are presented in Figure 11.

From the results, it was noticed that the addition of coir fibers in mortars, considering these two ratios parameters, has shown a more ductile behavior and less susceptibility to cracking, regardless of the type of binder used. Moreover, it is clear, as expected, that the mortars with cement and air-lime exhibited higher ductility when compared to the cement-based mortars.

Taking into account the fracture toughness values, it can be seen that the modified mortars presented an increase in relation to the REF’s. For cement-based mortars, shorter fibers seemed to be more effective in improving the fracture toughness of the mortars. C 1.5-20c attained up to 100% higher toughness values when compared to the reference mortar. On the other hand, for mortars with cement and air-lime this increment was not so high, since the highest value was of C 3.0-20cl, which was 65% higher than that of REF 1:1:6. It was observed that the fracture toughness has increased as increasing the fiber content and fiber length.

To conclude, mortars’ toughness exhibited a remarkable enhancement when coir fibers were added. Moreover, according to the parameters analyzed, the coir fiber-reinforced mortars showed a more ductile behavior. Therefore, the mortars with incorporation of coir fibers revealed to be less susceptible to cracking. Furthermore, for cement-lime mortars, a higher volume fraction of fibers resulted in better load-carrying capacity after achieving the maximum peak load.

Other authors also reported this improvement in fracture toughness by adding vegetable fibers in mortars [14,31,34,35]. Xie et al. [36] found that the incorporation of 16 wt.% of rice and bamboo cellulosic fibers increased the fracture toughness in 37 and 45 times, respectively, compared to the mortar without fibers. Rupasinghe et al. [31] and Sathiparan et al. [14] found that the incorporation of coir fibres in cement-lime mortars presented a better performance in terms of fracture toughness, residual strengths, and ductility. The authors reported that the fracture toughness increased with an increase in the volume fraction of coir fibers. The sample with a higher fibers content has increased about 10 times over the control mortar [31].

From the load-deflection curves of the specimens, it can be seen the load-carrying capacity after the maximum peak load achieved. It was observed that the incorporation of coir fibers enhanced the post-peak behavior of the mortars. The cement-based mortars with coir fibers showed a lower decay of load after the mortar achieved its maximum force, since the reference mortars exhibited a brittle failure mode. It is evidenced that the coir fiber-reinforced mortars with cement and air-lime withstand a residual load during failure and sustain greater deformations. It is stressed that a higher volume fraction of fibers presented a better load-carrying capacity.

Other authors also analyzed the load-deflection curves of vegetable fiber-reinforced mortars and verified a similar trend obtained in this work [1,14,35,37]. Hwang et al. [1] reported that the area under the load-deflection curves of the coir fiber-reinforced cement mortars was greater than that of the mortar without fibers, which indicates an increase in toughness indices of the modified mortars. Xie et al. [35] and Benaimeche et al. [37] also followed a similar trend, showing that the addition of vegetable fibers in cement mortars, namely rice, bamboo, and date-mesh palm fibers improved the post-peak behavior by providing a not abrupt failure. This effect was attributed to the bridging mechanism of the fibers across the cracks, which enable the mortars to distribute the stresses in non-brittleness mode [14]. In the study of Pereira et al. [34], it was found that longer sisal fibers exhibited higher fracture toughness than the shorter ones.

It can be concluded that the binders used to produce rendering mortars with coir fibers addition affects their properties. In order to the fibers to be effectively requested to improve the mortars behavior in terms of failure, the modulus of elasticity of the fibers must be compatible with the modulus of elasticity of the mortar. Therefore, it is important to note that the modulus of elasticity of the coir fibers is more similar to the modulus of elasticity obtained in cement and air-lime mortars. For this reason, it is clear that the cement-lime mortars presented a better cracking behavior with coir fibers addition. It is also stressed that the volume fraction of the fibers has more influence than the fibers’ length in improving specific properties, namely cracking susceptibility. In order to evaluate the durability of the coir fibers inside the matrix, the tests were performed over time. From the results, it was clear that the improvement of the fiber’s addition in mortar’s mechanical properties slightly decayed, over time. However, in general, the fibers improved the failure mode of the mortar, increasing their ductility that remains over time.

### 3.7. Open Porosity

The results of the open porosity test are presented in Figure 12. This test measures the volume of interconnected pores inside the mortars, and it is strongly correlated to the modulus of elasticity, ultra-sound pulse velocity, and mechanical strengths. From the results, it can be noticed that the incorporation of coir fibers increased the porosity of the mortars, regardless of the type of the binder used. At 28 days, the C 1.5-20cl and C 3.0-20cl both presented a reduction of the modulus of elasticity of about 16%, and an increase of the open porosity of 25% and 27%, respectively, relative to the reference mortar.

Mortars with longer and higher volume fraction of fibers exhibited higher porosity. At 365 days, C 3.0-20c and C 3.0-20cl increased 15% and 4% over their respective reference mortar. This increase in open porosity of the mortars with fibers is attributable to the fiber-matrix interfacial bond, which may generate some voids inside the matrix due to the fibers clustering. Previous studies are in agreement with those results found in this work [1,14,34]. The incorporation of coir fibers increased the porosity of the mortars. It can be seen that the mortars with higher porosity obtained the lower modulus of elasticity.

## 4. Conclusions

From the results obtained in this study, the following conclusions can be drawn:Coir fiber addition reduces the mortars’ workability, regardless of the type of binder used. As increasing the fiber length and volume fraction, a higher mixing water content is required to achieve the intended consistency when compared to the reference mortars.The mortars with coir fibers presented a more ductile behavior and less susceptibility to cracking than that of the control mortars, since they presented lower modulus of elasticity and higher fracture toughness. The addition of coir fibers also increased the porosity of the mortars due to the fibers’ clustering inside the matrix.Concerning the mechanical behavior of the mortars, the coir fiber addition improved in the first ages of the mortars. Over time, the coir fibers did not significantly affect their compressive and flexural strengths.

The findings of this current work show that the addition of coir fibers in rendering mortars led to improvements mainly in terms of cracking behavior. Furthermore, it provided an environmentally-friendly and low-cost product. The characterization of the coir fiber-reinforced mortars highlighted that the volume fraction of the fibers and the binder used are the most influencing factors to improve the brittleness of the mortar. A higher fiber content and cement-lime as a binary binder obtained the highest fracture toughness, according to the load-deflection curves.

## Figures and Tables

**Figure 1 materials-14-00823-f001:**
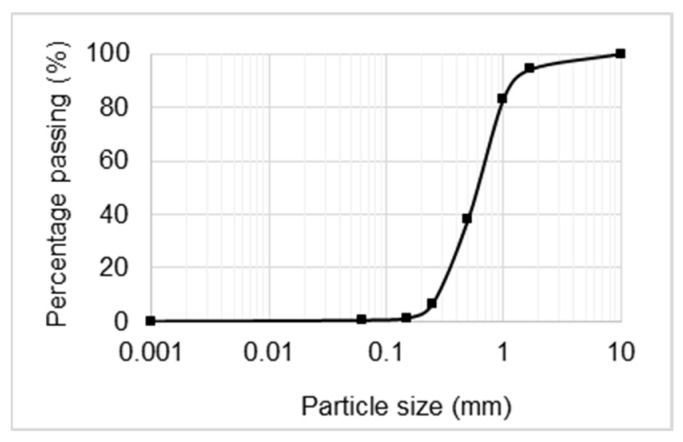
Size distribution of the sand used.

**Figure 2 materials-14-00823-f002:**
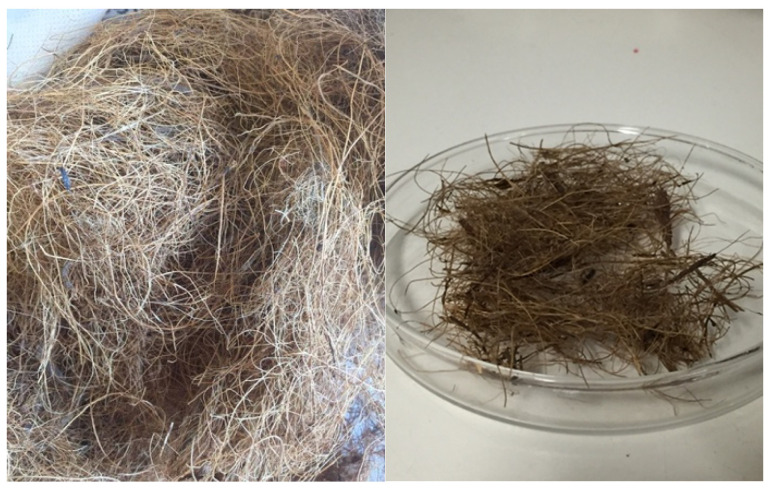
Coir fibers used in this work.

**Figure 3 materials-14-00823-f003:**
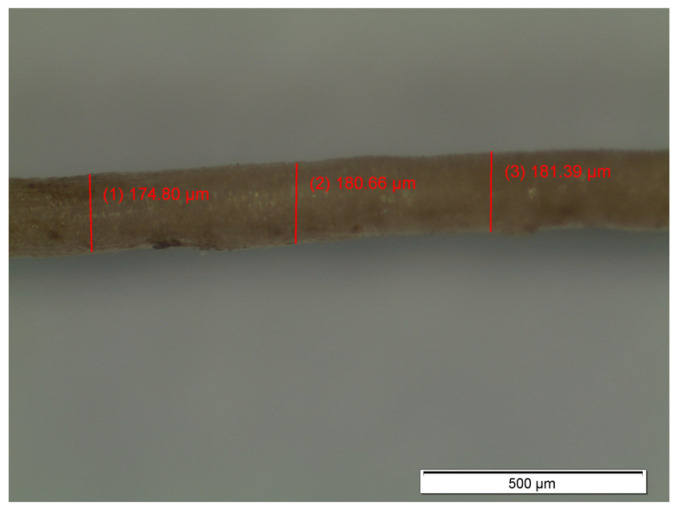
Optical microscope image of a coir fiber.

**Figure 4 materials-14-00823-f004:**
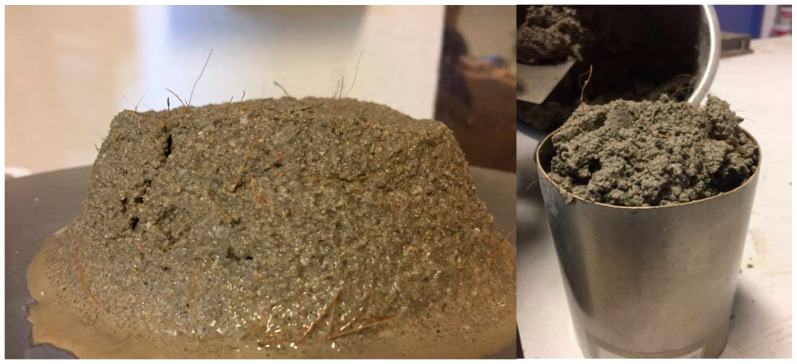
Flow table and bulk density test for coir fiber-reinforced cement mortar.

**Figure 5 materials-14-00823-f005:**
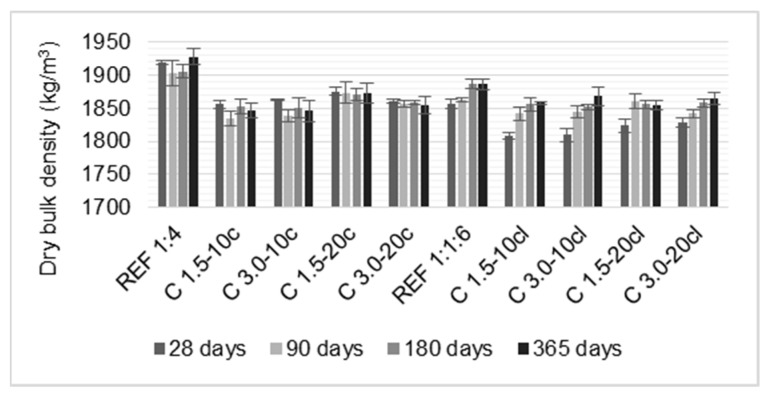
Dry bulk density of the hardened mortars.

**Figure 6 materials-14-00823-f006:**
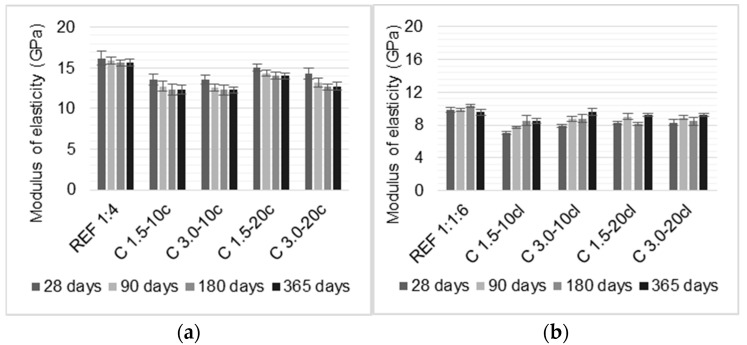
Dynamic modulus of elasticity of the: (**a**) cement mortars and (**b**) cement-lime mortars.

**Figure 7 materials-14-00823-f007:**
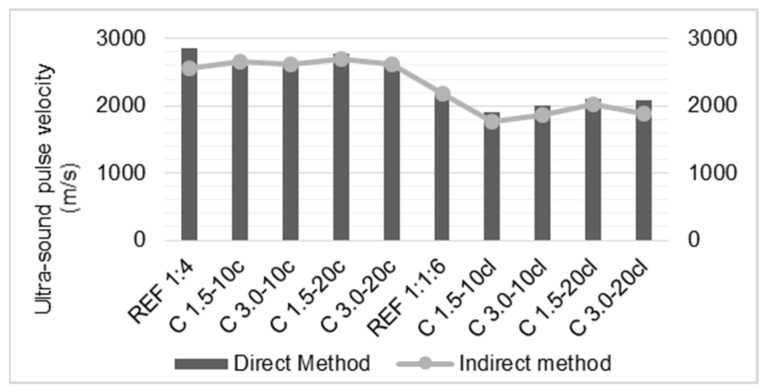
Ultra-sound pulse velocity of the mortars.

**Figure 8 materials-14-00823-f008:**
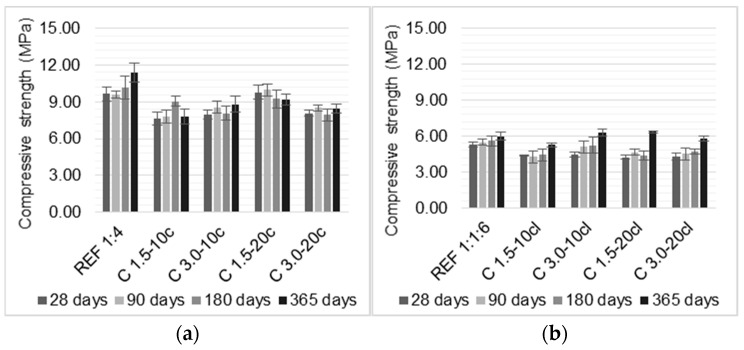
Compressive strength of the mortars: (**a**) cement mortars and (**b**) cement-lime mortars.

**Figure 9 materials-14-00823-f009:**
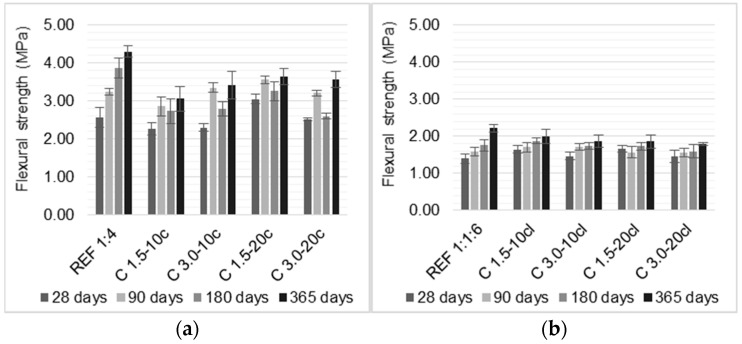
Flexural strength of the mortars: (**a**) cement mortars and (**b**) cement-lime mortars.

**Figure 10 materials-14-00823-f010:**
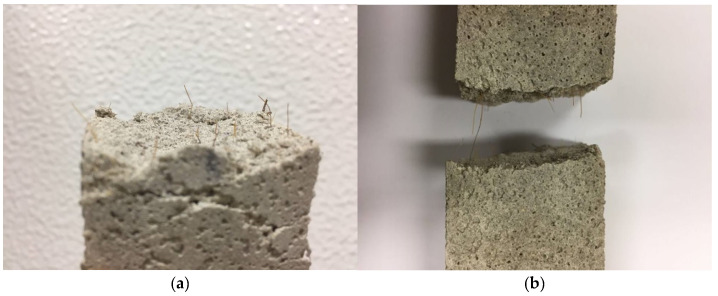
Coir fiber-reinforced cement-based mortars specimens after (**a**) compressive strength test and (**b**) flexural strength test.

**Figure 11 materials-14-00823-f011:**
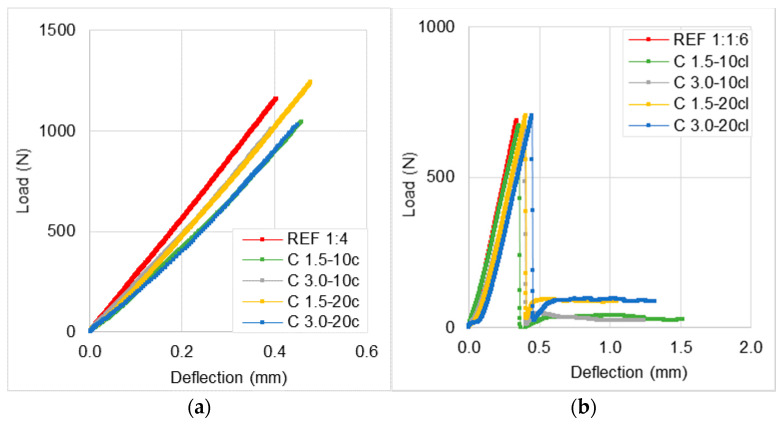
Load/deflection curves obtained from flexural strength of the mortars: (**a**) cement mortars and (**b**) cement-lime mortars.

**Figure 12 materials-14-00823-f012:**
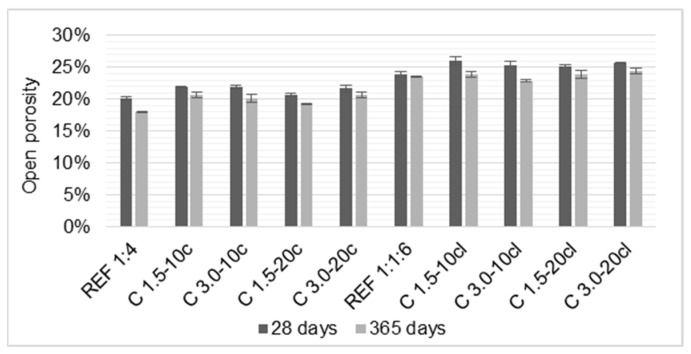
Open porosity test results of the mortars tested.

**Table 1 materials-14-00823-t001:** Bulk density of the constituents.

Component	Apparent Bulk Density (kg/m^3^)
Cement	975.5
Air-lime	565.7
Sand	1230.8
Coir 1.5 cm	5.4
Coir 3.0 cm	2.6

**Table 2 materials-14-00823-t002:** Composition of the mortars mixes by mass.

Mortar	Water (mL)	Cement (g)	Air-Lime (g)	Sand (g)	Coir Fiber (g)	Incorporation
REF 1:4	445	487.8	-	2461.6	0	0%
C 1.5-10c	415	439.1	-	2215.4	1.4	10% of 1.5 cm
C 3.0-10c	430	439.1	-	2215.4	0.7	10% of 3.0 cm
C 1.5-20c	370	390.2	-	1969.3	2.7	20% of 1.5 cm
C 3.0-20c	400	390.2	-	1969.3	1.3	20% of 3.0 cm
REF 1:1:6	465	304.8	176.8	2307.8	0	0%
C 1.5-10cl	425	274.4	159.1	2077.0	1.4	10% of 1.5 cm
C 3.0-10cl	420	274.4	159.1	2077.0	0.7	10% of 3.0 cm
C 1.5-20cl	396	243.9	141.4	1846.2	2.7	20% of 1.5 cm
C 3.0-20cl	396	243.9	141.4	1846.2	1.3	20% of 3.0 cm

**Table 3 materials-14-00823-t003:** Fresh mortars properties.

Mortar	Water/Binder Ratio	Bulk Density (kg/m^3^)
REF 1:4	0.91	2005 ± 4
C 1.5-10c	0.94	1959 ± 16
C 3.0-10c	0.97	1971 ± 5
C 1.5-20c	0.94	1940 ± 30
C 3.0-20c	1.02	1989 ± 15
REF 1:1:6	0.98	1999 ± 8
C 1.5-10cl	0.99	1989 ± 18
C 3.0-10cl	0.98	2000 ± 7
C 1.5-20cl	1.03	1986 ± 8
C 3.0-20cl	1.03	1993 ± 8

**Table 4 materials-14-00823-t004:** Parameters related to cracking susceptibility of the mortars tested.

Mortar	Dynamic Modulus of Elasticity (MPa)	Flexural Strength (MPa)	Compressive Strength (MPa)	E/σ_f_	σ_f_/σ_c_	Fracture Toughness (N·mm)
REF 1:4	16,210	2.56	9.66	6332	0.27	195
C 1.5-10c	13,560	2.27	7.64	5974	0.30	225
C 3.0-10c	13,740	2.31	7.22	5948	0.32	192
C 1.5-20c	15,010	3.06	9.78	4905	0.31	393
C 3.0-20c	14,360	2.52	8.09	5698	0.31	217
REF 1:1:6	9820	1.40	5.27	7014	0.27	135
C 1.5-10cl	7020	1.65	4.35	4255	0.38	155
C 3.0-10cl	7860	1.47	4.44	5347	0.33	149
C 1.5-20cl	8240	1.67	4.24	4934	0.39	212
C 3.0-20cl	8290	1.47	4.30	5639	0.34	223

## Data Availability

The data presented in this study are available on request from the corresponding author.

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
