# Peer review of "Physical and Mechanical Performance of Coir Fiber-Reinforced Rendering Mortars"

_materials, 2021, doi:10.3390/ma14040823_

Round 1
Reviewer 1 Report
Suggestions to the Authors
The main focus of work is to investigate the feasibility of the cement and cement-lime render mortars with the addition of coir fibres, in order to minimise cracking by improving the mortars’ ductility. The physical-mechanical properties of coir fibres reinforced mortars’ with different compositions were investigated at fresh and hardened state through several tests.
The topic dealt is interesting and the work is well structured, but requires substantial clarification regarding the use of these mortars as renders/plasters. In fact, generally the best binders for rendering mortars are not cements (more used in concretes), but mortars based on hydraulic lime (possibly natural, NHL) or suitable mixes of lime and cement. This latter mixing was used for about half of the specimens in the research presented here, together with specimens made with only cement as a binder. In this last case the physical-mechanical behaviour, even if it shows good results, involves other well-known problems when used as renders/plasters. In fact, with a highly hydraulic binder such as cement, a sufficient vapour permeability is not achieved, a property now required by numerous construction standards. Undoubtedly, the use of coir fibres reduce the high rigidity of cement mortars, giving greater toughness to the deformation imposed by the compressive and flexural stresses. However, mortars with mixed composition (cement + hydrated lime, tested in the research) or with hydraulic limes (not tested) are probably more suitable. Probably, the cementitious mortars reinforced with coir fibres can be used in particular cases where high permeability is not required, such as: i) specific waterproof plasters, ii) bedding mortars, iii) high resistance and hydraulic screeds for floorings.
Having made these considerations, I ask you to deepen these arguments in the manuscript so that you can clearly understand what the intent of the research is and what are the practical uses of these types of tested mortars.
Thus, the work is likely to be published, but at the moment, it needs a major revision of manuscript. Also some attached figures and contents of tables must be review with a new and homogenised re-presentation format. Several corrections are highlighted in the attached pdf file.
Below are the general suggestions for the manuscript and more, in addition to those already highlighted in the attached pdf.
Introduction
We ask to better combine the bibliography to which the text refers in the description of the investigations already carried out and results obtained by other authors on the similar analysed reinforced mortars.
Furthermore, at the end of the text of this section where the aims of the work are described, especially in relation to the possible use of reinforced mortars on the civil construction.
Methods
The instrumental and method standards are described in correct way, only some corrections are present (see pdf file). In addition, some sentence present in other text sections (e.g., Results) must be inserted in this section.
To understand the physical behaviour of tested mortars I suggest to insert in this section the description of mineralogical composition (for example by XRD analysis) of the sand used as aggregate in the mortars and the composition of coir fibres. This latter aspect is important to define the possible reaction with the cement binder of coir fibres and/or their chemical-physical decay.
Results
The results are well presented, but it would be better to treat the data of the various properties analysed with a greater connection on the interpretative level. At the moment they are too split. If you prefer, alternatively, you can insert a further paragraph (Discussion) immediately after the Results section, also drawing inspiration from what is written in the conclusions which are well described but are a bit too long.
In the discussion it is necessary to deal the results highlighting especially the relationships between the compositional, physical and mechanical characteristics of reinforced mortars. Moreover, it is important to explain a possible chemical-physical decay of coir fibres over time (within 365 days) that lead to a their micro-disintegration.
Conclusions
The conclusions are clear and well described; however, also considering the length of the text, they seem more like a discussion of the results. It is necessary to evaluate whether it is good to leave the conclusions as they are or to transfer the content of this section to the new section of the discussion of the results, leaving space in the conclusions for more practical suggestions on the real use of such reinforced mortars in the building sector.
Figures and tables
It is necessary to make some corrections, especially of a graphic nature (see the suggested corrections within the pdf file).

Reviewer 2 Report
In general, the paper is well written and it can contribute to the practice and literature. The reviewer would like to give some minor comments as follows:
- Why the author decided to add 10% and 20% coir fibre by total mortar volume.
- Why the author intentionally cut the coir fibre with the length of 1.5 and 3.0 cm, the author should give some comments on these values.
- Coir fibre is an organic material, it can be degraded with time when exposing to wet condition or other conditions. How does the author consider this problem?
- The citation of reference in the pdf file that the reviewer received has many errors. Please kindly check it.
Regards,
Reviewer 3 Report
Interesting article.
Can you comment on the method used to keep precise control of fiber length? I would have liked seen a fiber size distribution value in each case. Also, with aid of a microscope, what would be an average diameter for these fibers? what is the Length/diameter ratio or aspect ratio. The aspect ratio of fillers plays an important role in reinforced composite materials, similar concept here.
Also, since these fibers are entangled, how did the authors control fiber clustering when mixing and making the mortar?
In the methods, what ratio between components is kept fixed?, as you add coir fiber one of the components is reduced to keep same proportions, how does that affect the results?
Adding more water to improve workability when using coir fibers changes cement hydration and ultimately affects the final properties, can you comment on this and the effect on properties.
Addition of fibers would help some properties and hurt others, what is your comment about the maximum amount of fiber content to add to see that "critical" point where fiber addition no longer helps?
Round 2
Reviewer 1 Report
-